# Breaking the Chain: Protease Inhibitors as Game Changers in Respiratory Viruses Management

**DOI:** 10.3390/ijms25158105

**Published:** 2024-07-25

**Authors:** Christos Papaneophytou

**Affiliations:** Department of Life Sciences, School of Life and Health Sciences, University of Nicosia, Nicosia 2417, Cyprus; papaneophytou.c@unic.ac.cy; Tel.: +357-22841941

**Keywords:** respiratory viruses, proteases, inhibitors, human rhinovirus, adenovirus, SARS-CoV-2

## Abstract

Respiratory viral infections (VRTIs) rank among the leading causes of global morbidity and mortality, affecting millions of individuals each year across all age groups. These infections are caused by various pathogens, including rhinoviruses (RVs), adenoviruses (AdVs), and coronaviruses (CoVs), which are particularly prevalent during colder seasons. Although many VRTIs are self-limiting, their frequent recurrence and potential for severe health complications highlight the critical need for effective therapeutic strategies. Viral proteases are crucial for the maturation and replication of viruses, making them promising therapeutic targets. This review explores the pivotal role of viral proteases in the lifecycle of respiratory viruses and the development of protease inhibitors as a strategic response to these infections. Recent advances in antiviral therapy have highlighted the effectiveness of protease inhibitors in curtailing the spread and severity of viral diseases, especially during the ongoing COVID-19 pandemic. It also assesses the current efforts aimed at identifying and developing inhibitors targeting key proteases from major respiratory viruses, including human RVs, AdVs, and (severe acute respiratory syndrome coronavirus-2) SARS-CoV-2. Despite the recent identification of SARS-CoV-2, within the last five years, the scientific community has devoted considerable time and resources to investigate existing drugs and develop new inhibitors targeting the virus’s main protease. However, research efforts in identifying inhibitors of the proteases of RVs and AdVs are limited. Therefore, herein, it is proposed to utilize this knowledge to develop new inhibitors for the proteases of other viruses affecting the respiratory tract or to develop dual inhibitors. Finally, by detailing the mechanisms of action and therapeutic potentials of these inhibitors, this review aims to demonstrate their significant role in transforming the management of respiratory viral diseases and to offer insights into future research directions.

## 1. Introduction

Respiratory viruses represent a significant global health burden, accounting for over 3.9 million deaths each year and positioning respiratory viral infections (VRTIs) among the top five causes of mortality worldwide [1]. The range of common respiratory pathogens includes influenza virus, human rhinovirus (HRV), coronavirus (CoV), respiratory syncytial virus (RSV), parainfluenza (PIV), human metapneumovirus (HMPV), adenovirus (AdV), and bocaviruses. These pathogens are notably active during the colder seasons of fall and winter, leading to widespread illnesses such as the flu, the common cold, and severe respiratory syncytial virus infections, particularly in vulnerable populations like infants and the elderly [2].

In less developed areas, the impact of VRTIs is especially severe, being the foremost cause of mortality in children under five years of age. The high case-to-fatality ratios in these regions, compared to temperate areas, underscore the critical need for effective public health interventions and access to medical care. Among the spectrum of respiratory diseases, lower respiratory tract infections rank as the third leading cause of global mortality, but in low-income countries, they ascend to the leading cause. Despite the prominence of pneumonia, other respiratory conditions such as asthma and chronic obstructive pulmonary disease (COPD) contribute significantly to morbidity and mortality but are often under-reported and less emphasized in public health data [3,4].

The transmission dynamics of these viruses involve the expulsion of respiratory droplets by infected individuals through coughing or sneezing. These droplets can settle on surfaces where the virus remains viable or remain airborne in smaller droplet nuclei (<5 μm), which can penetrate the lower respiratory tract. The viral entry into host cells begins with the attachment to specific cell-surface receptors, followed by internalization through mechanisms like endocytosis and subsequent release of viral genetic material into the host cell, initiating replication [5].

The recurrent, seasonal outbreaks of respiratory viruses cause not only acute illnesses but also precipitate severe economic impacts because of healthcare costs and lost productivity. These seasonal epidemics particularly affect susceptible groups, including the elderly, those with compromised immune systems, and individuals with underlying health conditions. The recent global spread of coronavirus disease 19 (COVID-19) highlighted the profound societal and health consequences of respiratory viruses, with severe-acute-respiratory-syndrome-related coronavirus-2 (SARS-CoV-2) causing extensive morbidity and mortality and stressing healthcare systems across the world [6].

Beyond the acute manifestations, respiratory viruses also exacerbate chronic health conditions. For example, viral infections are responsible for about 50% of exacerbations in COPD patients, illustrating the interconnected nature of acute viral infections and chronic respiratory diseases [7]. This relationship underscores the extensive burden these viruses place on both individual health and broader public health systems [8].

Addressing these challenges is complicated by the high mutation rates of respiratory viruses, which facilitate frequent antigenic shifts and drifts. This genetic variability not only hampers vaccine development but also diminishes the effectiveness of antiviral drugs [9]. Traditional antiviral treatments often struggle with issues such as low solubility, poor stability, significant side effects, and the quick emergence of resistance [10]. Moreover, the ability of viruses to alter host metabolic pathways adds an additional layer of complexity to managing these infections, causing more innovative therapeutic strategies [11].

Viral proteases are vital targets for the development of new drugs due to their indispensable role in the life cycle of viruses [12]. These enzymes are found in many pathogenic viruses including those affecting the upper and/or lower respiratory tract, such as RVs, AdVs, and CoVs. They are crucial for processing the viral polyprotein into its functional components, which includes the ability to self-cleave. Typically, these viruses synthesize their proteins within a large polyprotein that includes one or more proteases. These proteases can activate themselves and subsequently divide the rest of the polyprotein into active proteins [13]. The inhibition of these proteolytic activities can halt the production of infectious viral progeny and mitigate disease severity, underscoring the potential of structure-based drug design [13,14]. Proteases exhibit distinct mechanisms and specificities across different viral families, making them diverse and strategic targets for antiviral treatment. This diversity in proteolytic mechanisms is evident when comparing viruses within the same category affecting the upper respiratory tract. For instance, influenza viruses, which belong to the Orthomyxoviridae family, utilize a different proteolytic approach [15,16]. Their replication depends on proteases that cleave the hemagglutinin (HA) protein, a critical step for the virus to enter host cells. Unlike the self-cleaving proteases of RVs and CoVs, the HA protein in influenza must be cleaved by host cell proteases, a necessity for merging the viral envelope with the host cell membrane and allowing the viral genome to initiate replication [17]. This fundamental difference in protease activity, where host proteases rather than viral ones are critical, highlights the variation in antiviral drug targets and treatments among different viral infections. This variability underlines the importance of developing a broad spectrum of antiviral strategies to cater to different viral mechanisms [18]. The use of protease inhibitors has indeed been a significant development in the management of respiratory viruses, particularly during the COVID-19 pandemic [19].

This review aims to explore the role of protease inhibitors in managing infections from respiratory viruses that depend on proteases for their maturation, using RVs, AdVs, and SARS-CoV-2 as examples. Despite the significant disease burden from common cold viruses, research on protease inhibitors for these pathogens is limited, particularly compared to the extensive focus on SARS-CoV-2’s main protease. A search in PubMed with the keywords “SARS-CoV-2”, “main protease”, and “inhibitors” yields over 1500 publications, whereas similar searches for HRV and AdV proteases return fewer than 100 papers each. This disparity highlights an opportunity to deepen the investigation into lesser-studied viral proteases. This review also discusses the mechanisms of these viral proteases, examines the current landscape of protease inhibitors, and identifies gaps in the literature. By leveraging existing data and exploring new compounds, it is suggested that repurposed inhibitors could revolutionize the treatment and prevention of respiratory viral diseases. An initial discussion on HRV, AdV, and SARS-CoV-2 sets the stage, followed by an analysis of their viral proteases and concluding with an overview of current research efforts to identify effective inhibitors.

## 2. Respiratory Viruses Dependent on Proteases for Replication

As mentioned previously, several viruses are common respiratory pathogens (reviewed in [20]); however, the most significant viruses affecting the respiratory tract that employ proteases for their replication—which have been exploited to design drugs with high therapeutic value—are, primarily, picornaviruses, AdVs, and CoVs. The key characteristics of these viruses are summarized in Table 1, and a brief discussion of these viruses and their proteases is provided in the subsequent paragraphs.

### 2.1. Human Rhinoviruses

RVs are small, non-enveloped viruses with a positive-sense, single-stranded RNA [(+)ssRNA] genome, classified under the Picornaviridae family. Human RVs are divided into three species (A, B, and C), encompassing over 150 identified serotypes [21]. RVs typically infect the upper respiratory tract and are a primary cause of the common cold. However, they can also infect the lower respiratory tract and have been linked to more severe symptoms, including asthma exacerbations, bronchiolitis, pneumonia, and chronic obstructive pulmonary disease (COPD) [22]. Currently, there are no specific antiviral treatments or vaccines available for RVs.

### 2.2. Adenoviruses

AdVs are nonenveloped DNA viruses from the distinct Adenoviridae family and play a significant role in respiratory tract diseases among both pediatric and adult populations [23]. These viruses are notable for their lack of a host-derived envelope, which not only increases their resistance to detergents and alcohol-based sanitizers but also complicates infection-control efforts, particularly during epidemics [24]. The Adenoviridae family encompasses six subgroups—A, B, C, D, E, and F—with a total of 51 identified serotypes [24,25]. These serotypes are responsible for a range of illnesses, from mild upper respiratory infections and bronchitis to severe, life-threatening multiorgan diseases. Typically, adenoviral infections are self-resolving; however, approximately 5% to 10% of all lower respiratory tract infections in pediatric patients are attributed to adenovirus [24]. Certain serotypes, notably 11 and 35, have been associated with severe outcomes like deadly adenoviral pneumonia in newborns, as documented in several cases studies [26]. Furthermore, the clinical presentation of adenoviral infections can vary significantly among immunocompetent adults, underscoring the diverse pathogenic potential of different serotypes [27]. Beyond humans, adenoviruses also infect a wide range of other vertebrates, highlighting their broad biological impact [28]. This variability in disease manifestation and the widespread nature of adenoviral infections emphasize the challenges in managing and controlling these pathogens, particularly in settings prone to outbreaks.

### 2.3. Coronaviruses

SARS-CoV-2, the causative agent of COVID-19, belongs to the Coronaviridae family and Coronavirinae subfamily [29]. These viruses are large, enveloped, single-stranded RNA viruses, ranging from 65 to 125 nm in diameter [30]. Members of the Coronavirinae subfamily, which infect a variety of mammals, are categorized into four sub-groups based on sequence homology: alpha, beta, gamma, and delta [31]. To date, seven CoVs are known to infect humans; four of these (two α-CoVs and two β-CoVs) typically cause mild cold-like symptoms, while three, including SARS-CoV, Middle East respiratory syndrome coronavirus (MERS-CoV), and SARS-CoV-2, can lead to severe or fatal outcomes. SARS-CoV first emerged in 2002, followed by MERS-CoV in 2012. Most recently, SARS-CoV-2 appeared in December 2019, triggering the ongoing global pandemic [32]. CoVs infect both the upper and lower respiratory tracts. While infections often result in only mild symptoms, they can also lead to more severe conditions, such as viral pneumonia and exacerbated COPD [33]. The range of outcomes from COVID-19 varies greatly, from asymptomatic cases to severe, life-threatening conditions with the potential for long-term health effects [34]. Before the SARS-CoV-2 outbreak, there were no approved vaccines for any coronavirus. However, the urgency of the pandemic accelerated vaccine research and development, resulting in the historic approval of the first mRNA vaccine at the end of 2020 [35].

## 3. Viral Proteases: The Ultimate Tool for Viral Maturation and Invasion

Proteases, also known as peptidases or proteinases, are enzymes that catalyze the cleavage of peptide bonds, primarily through hydrolytic mechanisms [36]. These enzymes are integral to various biological processes, including protein processing, catabolism, blood coagulation, immune response, cell signaling, and apoptosis [37]. Ubiquitous in all living organisms, proteases are also encoded by many viruses, such as retroviruses, herpesviruses, flaviviruses, and coronaviruses, playing crucial roles in their life cycles [38].

Viral proteases cleave large polyproteins encoded by the viral genome into functional proteins essential for the virus’s replication and maturation. This process is vital for proper capsid assembly and the production of infectious particles, enabling the virus to hijack the host cell’s machinery for its own proliferation. The activity of viral proteases is characterized by a nucleophilic attack on the carbonyl carbon of the peptide bond, facilitated by nucleophiles generated from the side chains of serine, cysteine, or from a water molecule coordinated to aspartic acid [15].

The mechanism involves a catalytic dyad or triad typically comprising an acidic residue such as aspartic acid, an alkaline residue like histidine, and a nucleophilic residue, either serine or cysteine. Proteases are classified based on the nature of the catalytic residue involved in types such as serine, threonine, cysteine, aspartic, glutamic, and metalloproteases, or based on structural features like the chymotrypsin-like fold, characterized by a two-β-barrel motif [39]. The catalytic triads of HRV–3C protease (HRV–3C^pro^) and adenovirus protease (AVP), as well as the catalytic dyad of SARS-CoV-2 main protease (M^pro^), are presented in Figure 1. A brief discussion on the structure, function, and specificity of each protease is provided in the following paragraphs.

Despite structural similarities with eukaryotic proteases, viral proteases have unique substrate specificities and regulatory mechanisms that differ significantly across viral families. They can recognize and cleave diverse substrate sequences at different rates in a highly ordered sequence, making them ideal targets for therapeutic intervention. Inhibition of these proteases can effectively block the production of infectious progeny and mitigate viral pathogenicity [40].

Clinically important inhibitors have been developed for several viral proteases, notably those of human immunodeficiency virus (HIV) and hepatitis C virus (HCV). Recent approvals of inhibitors targeting the main protease of SARS-CoV-2 underscore the ongoing relevance and potential of protease inhibitors in antiviral therapy. Continued research and development of these inhibitors are critical given their crucial role in viral replication and the pathogenesis of infectious diseases [13].

In the following paragraphs the main characteristics of HRV–3C^pro^, AVP, and SARS-CoV-2 M^pro^ are discussed.

### 3.1. Rhinoviruses Proteases

The genome of rhinoviruses, consisting of approximately 8000 bases, encodes a single polyprotein which is processed cotris anslationally by proteases such as 3C, 2A, and leader proteases [41,42]. The majority of this processing is executed by the 3C protease (3C^pro^), although 2A and leader proteases are found in only some picornaviruses. The 3C and 2A proteases are classified as cysteine proteases and share a Gly–X–Cys–Gly motif, akin to the Gly-Asp-Ser-Gly active site found in chymotrypsin-like serine proteases [43]. Crystal structures of these proteases in picornaviruses and other viruses like HRV 2 and 14 further confirm their similarity to chymotrypsin-like serine proteases [44,45]. For instance, the 3C protease of HRV14 comprises 182 residues, whereas the 2A protease of HRV2 consists of 142 residues [44]. Specifically, the active site of the 3C^pro^ in HRV14 includes a catalytic triad of Cys146, His40, and Glu71, which are positioned as in serine proteases [46]. The 2A proteases of HRV2 feature a catalytic triad of Cys106, His18, and Asp35.

### 3.2. Adenovirus Protease

AVP, also known as adenain, is a cysteine protease encoded by adenoviruses during the late phase of infection. This enzyme plays a critical role in the maturation of the virus and the production of infectious progeny virions [47]. AVP is characterized by a unique structure with a central mixed five-stranded β-sheet flanked by helices on both sides, closely resembling the structure of papain and Picornavirus leader protease in its overall organization. Despite some differences in the primary sequences, the structural conservation of the active site, including the catalytic triad (Cys122–His54–Glu71) and the oxyanion hole (Gln115), suggests a similar catalytic mechanism to papain [48]. AVP functions as a highly basic protein that specifically recognizes sequence motifs (M/I/L) XGX-G and (M/I/L) XGG-X, where X can represent any amino acid [49,50]. This protease is also noted for being phosphorylated and typically has 7–50 copies packaged in mature virions [51]. Synthesized in an inactive form, AVP becomes partially active upon packaging in immature virions and upon binding to viral DNA [51]. The activation of AVP occurs through its interaction with sliding pVI, which cleaves it at both N- and C-terminals to release an 11 amino acid peptide co-factor pVIc. This binding enhances AVP’s activity, enabling it to slide along the viral DNA within virions [52]. Moreover, AVP is essential not only for cleaving adenoviral precursor proteins such as IIIa, VI, VII, VIII, Mu/X, and TP within the virion but also for the proper disassembly and release of incoming virus particles in the cytoplasm [53]. Interestingly, AVP can cleave proteins even in the absence of viral DNA and pVIc, demonstrating flexibility in its activity [54,55]. For example, AVP encoded by different adenovirus strains, such as BAdV-3, HAdV-5, or porcine AdV-3, recognizes and cleaves non-consensus viral protease sites in various cellular contexts [54]. The availability of crystal structures of both the active and inactive forms of AVP has spurred the development and evaluation of anti-adenoviral drugs, highlighting the enzyme’s potential as a target for therapeutic intervention [56].

### 3.3. SARS-CoV-2 Main Protease

The SARS-CoV-2 main protease (M^pro^), also known as 3-chymotrypsin-like proteases (3CL^pro^), are essential cysteine hydrolases found in β-coronaviruses. These proteases play a crucial role in viral replication and are critical targets for the treatment and prevention of diseases like COVID-19 because of their role in virus maturation [57]. M^pro^ operates in a dimeric form, achieving full enzymatic activity only when two monomers, each composed of three domains (I, II, and III), join. Notably, domain III contains a helical segment that facilitates this dimerization process. The catalytic activity of M^pro^ is centered at the intersection of domains I and II, arranged into five sub-pockets (S1, S2, S3, S4, S5) that enhance its function [58,59]. It features a His45–Cys145 catalytic dyad, located between domains I and II, which is a structural motif shared with M^pro^ of other coronaviruses but different from the catalytic triad found in 3-chymotrypsin [60,61].

The M^pro^ is a conserved enzyme across all coronaviruses, and the amino acids in its substrates are sequentially numbered from the N-terminus to the C-terminus as -P4-P3-P2-P1 and P1′-P2′-P3′ [62]. The cleavage site is strategically positioned between P1 and P1′, where a glutamine residue is essential at the P1 position for enzymatic activity [63]. M^pro^ specifically recognizes its substrates via a consensus sequence of P2P1–P1′ and catalyzes the hydrolysis of the amide bond between P1 and P1′. In this sequence, P1 is invariably Gln, P1′ is either Ser or Ala, and P2 is a hydrophobic amino acid such as Leu, Phe, or Val. From a mechanistic perspective, the catalysis begins when the –SH group of Cys145, deprotonated by His41, attacks the carbonyl carbon of the amide bond between the P1 and P1′ residues. This attack results in the formation of a thioester intermediate. His41, now protonated, serves as an acid to facilitate the release of the P1′ amine. The subsequent hydrolysis of the thioester intermediate releases the P1 acid, thereby completing the catalytic cycle essential for viral maturation [64].

Upon viral entry, SARS-CoV-2’s genomic RNA is translated directly by the host ribosome into two large polyproteins. These are then processed by two key proteases—M^pro^ and the papain-like protease (PL^pro^)—into smaller components that are crucial for assembling new virions [65]. This process, alongside the RNA replication facilitated by RNA-dependent RNA polymerase (RdRp), is vital for viral replication, making these proteins primary targets for therapeutic interventions [66]. M^pro^, in particular, has been identified as the “Achilles’ heel” of the virus, making it a prime target for drug development efforts against SARS-CoV-2 [67].

Despite its similarities with proteases from other (+)-ssRNA viruses such as HIV and HCV, which have well-established inhibitors, effective approved therapies targeting SARS-CoV-2’s M^pro^ are still lacking [67] (discussed further below).

## 4. Current Status of Inhibitors of Viral Protease Affecting the Respiratory Tract

### 4.1. Inhibitors for the HRV–3C Protease: An Old Story

Most studies on discovering potential inhibitors of HRV–3C^pro^ were conducted between the late 1990s and early 2000s, with Agouron Pharmaceuticals (a Pfizer company) playing a significant role in this research area. Some examples of known HRV–3C^pro^ are illustrated in Figure 2. Initial attempts to discover HRV–3C^pro^ inhibitors have involved compounds which incorporate Michael acceptors into their molecular structure [68]. Such inhibitors typically form an irreversible, covalent adduct between the β-carbon atom of the unsaturated Michael acceptor moiety and the active site cysteine residue of their protease target [69]. The first report of HRV–3C^pro^ inhibitors containing Michael acceptors was provided by Kong et al. [70], detailing several substrate-derived, tetrapeptidyl compounds that incorporate C-terminal vinylogous esters. In brief, they synthesized peptidyl derivatives of vinylogous glutamine or methionine sulfone esters (e.g., Boc-Val-Leu-Phe-vGln-OR: Figure 2; compound **1**: R = Me; compound **2**: R = Et) and evaluated them as inhibitors of HRV–14 3C^pro^. Compounds **1** and **2**, along with several related tetra- and pentapeptide analogs, rapidly inactivated HRV–3C^pro^ with sub-micromolar IC_50_ values. An electrospray mass spectrometry confirmed the expected 1:1 stoichiometry of HRV–3C^pro^ inactivation by compounds **1**, **2**, and several other analogs. Additionally, compound **2** proved useful for active site titration of HRV–3C^pro^, which was previously unachievable due to the lack of a suitable reagent. In contrast, peptidyl Michael acceptors lacking a P4 residue exhibited greatly reduced or negligible activity against HRV–3C^pro^, consistent with previously established structure–activity relationships for HRV–3C^pro^ substrates. The hydrolysis of the P1 vinylogous glutamine ester to a carboxylic acid also significantly decreased inhibitory activity, aligning with the decreased reactivity of acrylic acids versus acrylic esters as Michael acceptors. Similarly, incorporating a vinylogous methionine sulfone ester in place of the corresponding glutamine derivative in compound **1** substantially reduced activity. Compounds **1** and **2**, along with several of their analogs, inhibited HRV replication in cell culture by 50% at low micromolar concentrations while exhibiting little or no cytotoxicity at 10-fold higher concentrations. Following the initial disclosure of the Michael acceptor-containing compounds, Agouron Pharmaceuticals released several patents and publications [71,72,73,74] describing their efforts to optimize a related series of 3C^pro^ inhibitors (reviewed in [68]).

In addition to the previously mentioned Michael acceptors, several other classes of HRV–3C^pro^ inhibitors have been reported. A series of compounds that combine a halomethyl carbonyl electrophile with an azaglutamine-containing peptidyl recognition element [75]. One representative compound (Figure 2, compound **3**) demonstrated potent, irreversible inhibition of the 3C^pro^ (k_inact_/K_i_ = 23,400 M^−1^s^−1^ for HRV-1B) and exhibited anti-rhinoviral activity in the cell culture (ID_50_ = 2.5 μg/mL, HRV-1B, MTC = 20.7 μg/mL). Several possible mechanisms for the irreversible inactivation of HRV–3C^pro^ by this compound and its analogs were proposed, all involving covalent modification of the enzyme.

Venkatraman et al. [76] reported another class of azaglutamine-containing HRV–3C^pro^ inhibitors. These compounds feature reactive ester-like functionalities instead of halomethyl carbonyl electrophiles and displayed moderate levels of 3C^pro^ inhibition. However, some members of this class were identified as slow-turnover substrates for the viral protease, complicating their ability to fully inhibit the enzyme. Additionally, the anti-rhinoviral properties of these compounds were not reported or examined. Finally, Agouron Pharmaceuticals described an unrelated series of substrate-derived tripeptides bearing C-terminal heterocyclic ketone moieties [77]. These compounds exhibited potent and reversible inhibition of HRV–3C^pro^ and demonstrated anti-rhinoviral activity in the cell culture. For instance, a benzothiazole-containing compound (Figure 2, compound **4**) showed strong reversible inhibition of HRV–14 3C^pro^ (K_i_ = 4.5 nM) and moderate antiviral activity in the cell culture (EC_50_ = 0.34 μM, CC_50_ = 250 μM). Although several other heterocyclic ketones were evaluated as HRV–3C^pro^ inhibitors, none displayed greater activity than the benzothiazole-containing compound **4**.

In another study, Jungheim et al. [78] reported the inhibition of HRV–3C^pro^ by homophthalimides. Kaldor et al. [79] synthesized and evaluated a series of peptide aldehydes as inhibitors of HRV–3C^pro^. These inhibitors, containing a C-terminal glutamine aldehyde, were prepared using a novel methodology involving the reduction in the corresponding glutarimide, which is easily derived from N-protected glutamine. Among the synthesized peptides, Boc-NH-Val-Leu-Phe-Gln-CHO (Figure 2, compound **5**) exhibited an IC_50_ of 0.6 μM against HRV–3C^pro^. Shepherd et al. [80] designed and synthesized small peptidic aldehyde inhibitors for human rhinovirus 3C protease. Di- and tripeptide aldehydes containing a methionine sulfone as a P_i_ surrogate for glutamine showed low micromolar enzyme inhibitory and antiviral tissue culture activity. Among these peptides, N-Cbz-Phe-Met(SO_2_)-CHO (LY338387; Figure 2, compound **6**) was identified as a potent reversible inhibitor of HRV–3C^pro^, with a K_i_ of 0.47 μM. It also exhibited good tissue culture activity (IC_50_ = 3.4 μM) without cytotoxicity (TC_50_ > 224 μM). Webber et al. [81] reported the design, synthesis, and biological evaluation of reversible, nonpeptidic inhibitors of HRV-3^pro^. A novel series of 2,3-dioxindoles (isatins; a representative example is shown in Figure 2, compound **7**) was designed using a combination of protein structure-based drug design, molecular modeling, and structure–activity relationship (SAR) studies. The C-2 carbonyl of isatin was envisioned to react with the cysteine at the active site of HRV–3C^pro^, forming a stabilized transition state mimic.

Dragovich et al. [72] reported the development of HRV–3C^pro^ inhibitors incorporating P1 lactam moieties instead of L-glutamine. These compounds, combining a tripeptidyl- or peptidomimetic-binding determinant with an ethyl propenoate Michael acceptor, form an irreversible bond with the enzyme’s active site cysteine. The lactam-containing inhibitors demonstrated superior HRV–3C^pro^ inhibition and antiviral properties compared to their L-glutamine-derived counterparts, exhibiting excellent selectivity and resistance to degradation by biological agents. One notable compound, AG7088 (rupintrivir; Figure 2, compound **8**), with an EC_90_ of approximately 0.10 µM across 46 serotypes, was highlighted for its potential and recommended for further preclinical development as an antiviral agent. The development of structural inhibitors, including rupintrivir and its analog AG7404, has progressed to clinical trials, showing effectiveness against various HRV strains [82]. Rupintrivir binds within the catalytic pocket of HRV type 15 3C^pro^, forming a covalent bond with Cys147, one of the key catalytic amino acids [82]. Furthermore, the EC_50_ value of rupintrivir against HRV type 14 3C^pro^ was reported as 52 nM [83]. Despite these advancements, rupintrivir failed to demonstrate sufficient efficacy in clinical trials [84,85].

Buthelezi et al. [86] identified two potential hits from the ZINC database, ZINC01537619 and ZINC601135028, which interact with the catalytic residues His40 and Cys147 of HRV–3C^pro^. Reich et al. [87] developed non-peptide benzamide-containing inhibitors targeting HRV–3C^pro^ through structure-based design, focusing on α,β-unsaturated cinnamate esters. These effectively inhibited HRV–3C^pro^ and exhibited antiviral activity with an EC_50_ of 0.60 μM in HRV-16-infected H1-HeLa cells. Despite potent inhibition, unsaturated ketones did not demonstrate antiviral activity in cellular assays and were reactive with nucleophilic thiols, limiting their therapeutic potential.

Natural naphthoquinone thysanone (Figure 2, compound **9**) [88,89] from *Thysanophora penicilloides* and its analogs, including (±)-9-deoxythysanone [90], have also emerged as potential inhibitors but have not proceeded to further evaluation. Recently, Jain et al. [91] utilized a Monte Carlo-based QSAR method to design new inhibitors, highlighting the importance of fingerprint-based drug design and Monte Carlo optimization in developing HRV–3C^pro^ inhibitors.

Our group [92] has recently highlighted the potential of phytochemicals as alternative inhibitors of HRV–3C^pro^. Through a two-step in silico screening of 2532 phytochemicals, we identified eight active compounds: apigenin, carnosol, chlorogenic acid, kaempferol, luteolin, quercetin, rosmarinic acid, and rutin. These candidates were subsequently evaluated in vitro. Molecular docking studies of the most promising candidates, carnosol and rosmarinic acid (Figure 2, compounds **10** and **10a**, respectively), revealed their competitive binding interactions with HRV–3C^pro^. Both compounds inhibited HRV–3C^pro^ activity by over 55% in vitro. Given their competitive binding to the enzyme’s active site, carnosol and rosmarinic acid warrant further investigation for their potential in developing treatments for the common cold.

Together, the above data reveal that research on identifying HRV–3C^pro^ inhibitors is relatively limited, with most studies conducted in the late 1990s and early 2000s. Despite causing primarily mild symptoms, the common cold remains a global health issue because of its widespread prevalence and significant impact on public health and economic productivity. Continued efforts to discover effective inhibitors for the HRV–3C^pro^ protease are essential to mitigate the persistent burden of this ubiquitous viral infection. The need for potential inhibitors of 3C^pro^ is urgent, as only a few inhibitors have been reported, and most of the studies are over 20 years old.

### 4.2. Adenoviruses Protease Inhibitors: A Short Story

As mentioned before, AdVs encode a cysteine endopeptidase, which is also called adenain and plays a crucial role in the viral lifecycle. This protease is essential for virion maturation and infectivity [93]. AVP is activated by two cofactors, namely pVIc [94] and viral DNA [95]. This activation is crucial for producing infectious viruses, making AVP a key target for antiviral therapy. McGrath et al. [96] determined the crystal structure of the AVP–pVIc complex at a resolution of 1.6 Å (PDB ID: 1NLN), which aids in the design of effective enzyme inhibitors. Although no specific anti-adenoviral drugs are currently approved, several studies reported potential inhibitors (reviewed in [97]). A brief discussion about the current status of adenain inhibitors is provided below, and examples of these inhibitors are presented in Figure 3.

Multiple reports indicate that piperazin derivatives inhibit AVP and reduce viral transcript levels and DNA replication. Based on the adenain–pVIC structure, McGrath et al. [56] reported on the development of first-generation inhibitors targeting the AVP. This research was part of a broader effort to develop antiviral agents specifically inhibiting AVP, employing computational docking and screening chemicals from the Open Chemical Repository of the NCI Developmental Therapeutics Program. Utilizing structure-based drug design, they focused on the active site and a conserved cofactor pocket revealed in the crystal structures of AVP. The lead compound identified, NSC 36806 (Figure 3), exhibited an IC_50_ of 18 μM, while this compound exhibits affinity for both AVP and AVP–pVIc. Given that a significant part of NSC 36806 did not interact with AVP and had a high molecular weight (829 g/mol), the authors initiated a substructure search within the NCI repository. This search identified eight structurally similar compounds, among which NSC 37248 and NSC 37249 were effective inhibitors (Figure 3). NSC 37248 was found to bind both the N-terminal binding pocket (NT pocket) and the active site in AVP–pVIc complexes, similar to NSC 36806. In contrast, NSC 37249 exclusively bound to the active site, acting as a competitive inhibitor. NSC 37249 was notably potent, with an IC_50_ of 140 nM against AVP and 490 nM in the presence of the cofactor.

Mac Sweeney et al. [98] reported a concise two-pronged hit discovery approach in which they identified tetrapeptide nitrile ((S)-2-((S)-2-(3-chlorophenyl)-2-(methylsulfonamido)acetamido)-N-(2-((cyanomethyl)amino)-2-oxoethyl)-3-phenylpropanamide) and pyrimidine nitrile (N-Benzyl-2-cyanopyrimidine-5-carboxamide) as complementary starting points for adenain inhibition (Figure 3, compounds **1** and **2**, respectively). X-ray co-crystal structures of adenain complexed with tetrapeptide nitrile and pyrimidine nitrile were obtained, and both inhibitors were found to be covalently bound to the catalytic Cys122 residue of adenain through their nitrile group. Although the tetrapeptide nitrile compound exhibited strong interactions with the protease and demonstrated high potency in the biochemical assay, it did not show activity in the viral replication assay, probably due its peptidic nature. In their quest to minimize the peptidic nature of the tetrapeptide nitrile, the researchers developed two non-peptide inhibitors: ((2S)-N-((2-cyanopyrimidin-4-yl)methyl)-2-(2-(3,5-dichlorophenyl)propanamido)-3-methoxypropanamide; and (2S)-N-((2-cyanopyrimidin-4-yl)methyl)-2-(2-(3,5-dichlorophenyl)propanamido)-4-hydroxypentanamide), achieving IC_50_ values in the picomolar range (Figure 3, compounds **3** and **4** respectively). In their continuous efforts to enhance the tetrapeptide nitrile, the same group focused on diminishing its peptidic characteristics by creating a novel substitution for the P4–P3 amide bond and exploring irreversible inhibitors [99]. The authors conducted a detailed investigation into tetrapeptide-nitrile-based derivatives, identifying a novel group of compounds that demonstrated efficacy against AVP. However, cytotoxicity may pose an issue with these compounds. Notably, there has been limited in vitro testing of these novel agents in cell cultures, and even less is known about their pharmacological properties in vivo.

More recently, Mali and Pandey [100] reported on 2D and 3D quantitative structure–activity relationships (QSAR) for a set of hydroxybenzamide analogs as potential inhibitors of human adenoviruses. This research utilizes molecular docking and QSAR modeling to identify compounds with significant inhibitory activity against these viruses. In silico docking and virtual screening studies revealed a higher binding affinity of dataset molecule 34 (−141.444 kcal/mol). The authors identified 12 best-ranked ZINC drug-like hits among a database of 10,639,400 molecules: ZINC04783387, ZINC14258364, ZINC02662530, ZINC14122355, ZINC60101163, ZINC02622531, ZINC01088642, ZINC08755993, ZINC60101204, ZINC02648703, ZINC03462578, and ZINC60101192. A re-docking of these top 12 ZINC hits on the 4PIE:3FO cavity revealed that ZINC01088642 had the highest docking score (−114.357 kcal/mol). In the in silico ADME/toxicity studies, molecule 34 and ZINC01088642 (Figure 3, compounds **5** and **6**, respectively) were found to be safe, with good intestinal absorption, aqueous solubility, medium blood–brain barrier (BBB) penetration, no eye corrosion, no skin irritancy, and non-mutagenic profiles.

Despite the health complications associated with adenovirus infections and the crucial role that viral protease plays in adenovirus survival and replication, research on identifying potential inhibitors of AVP remains limited. AdVs typically cause relatively mild diseases, which may explain the lower research priority given to these viruses compared to more lethal viruses [101]. Furthermore, the complexity of AVP’s structure and function presents significant challenges in the design and development of effective inhibitors. These factors combined have led to a paucity of focused efforts to develop targeted therapies against adenovirus proteases, underscoring a gap in antiviral research that could be critical for managing outbreaks, particularly in vulnerable populations or in settings where adenovirus transmission is high.

### 4.3. SARS-CoV-2 Main Protease Inhibitors: A Recent but Long Story

Drugs that specifically target and inhibit the SARS-CoV-2 M^pro^ offer promising alternatives in combating the COVID-19 pandemic. SARS-CoV-2 M^pro^ comprises 306 amino acids, forming three domains, with the active site nestled between domains I and II. Domain III plays a crucial role in dimerization, essential for M^pro^’s functionality [33]. As with the M^pro^ from SARS-CoV-1 and other coronaviruses, SARS-CoV-2 M^pro^ relies on two catalytic amino acids, His41 and Cys145, and a catalytic water molecule that forms a strong hydrogen bond with His41 [102]. Notably, Gly143 of SARS-CoV-2 M^pro^ is a prime residue for forming hydrogen bonds with ligands, followed by Glu166, Cys145, and His163 [103].

At the end of 2021, the FDA authorized nirmatrelvir (PF-07321332), a specific inhibitor of SARS-CoV-2 M^pro^, co-administered with the pharmacokinetic booster ritonavir (Paxlovid^®^) [104]. The development of nirmatrelvir was facilitated by prior efforts to target SARS-CoV, which emerged in 2002–2003. It should be noted that the SARS-CoV main protease is 96% identical to SARS-CoV-2 M^pro^.

Although allosteric binding sites in the SARS-CoV-2 M^pro^ structure have been reported [105], most inhibitors, including nirmatrelvir, target the enzyme’s active site [106]. Pfizer also developed an inhibitor of the SARS-CoV M^pro^, namely PF-00835231. This compound features an α-hydroxymethylketone warhead that forms a covalent, reversible bond with the protease’s catalytic cysteine [107]. By masking this reactive warhead with a phosphate group, the prodrug lufotrelvir (PF-07304814) was created, suitable for intravenous application due to its increased solubility compared to the parent compound. The phosphate group is cleaved by host alkaline phosphatase, which is abundant in the liver, lungs, and kidneys. Lufotrelvir has shown good activity against COVID-19, including several variant strains, in human clinical trials [108,109]. However, unlike nirmatrelvir, lufotrelvir is not orally active and must be administered via intravenous infusion. This requirement has made lufotrelvir a less favored candidate for clinical development overall [110].

Moreover, since the beginning of the COVID-19 pandemic, several efforts have been made to develop inhibitors for the SARS-CoV-2 protease. The development of M^pro^ inhibitors has been greatly facilitated by determining the crystal structures of viral proteases in complex potential inhibitors. These structures provide valuable insights into drug enhancement by tailoring inhibitors to match the structural dynamics—whether monomeric or dimeric—and the depth or breadth of the active site of the target enzymes. For example, AG7088 (rupintrivir) effectively inhibits rhinoviruses and other Picornaviral 3C-like proteases but not SARS-CoV M^pro^, which functions as a monomer with only two catalytic domains [111]. This underscores the need to adapt drugs based on sequence differences and structural specifics to enhance inhibitor specificity. Since the M^pro^ monomer is generally inactive, targeting the dimer form is optimal for drug development [112].

Among the initial inhibitors proposed was the N3 compound, which had previously been developed as a protease inhibitor for multiple coronaviruses, including SARS-CoV and MERS-CoV, as well as approved drugs like disulfiram and carmofur, and preclinical or clinical-trial drug candidates, such as ebselen, shikonin, tideglusib, PX-12, and TDZD-8 [102]. In the following years, a plethora of peptidic, peptidomimetic, and non-peptidic small molecule inhibitors have been developed targeting M^pro^. Thousands of compounds have since been proposed as SARS-CoV-2 M^pro^ inhibitors through computational methods like protein–ligand docking, high-throughput screening experiments, and computer-aided design and synthesis of new compounds. Many of these inhibitors incorporate an electrophilic warhead designed to covalently bond with Cys145. Extensive in vitro, in vivo, and in silico studies have demonstrated the effectiveness of these compounds in binding and inhibiting SARS-CoV-2 M^pro^.

Most of these inhibitors harbor an electrophilic “warhead” group specifically designed to react with and covalently bind to the nucleophilic –SH group of Cys145 [64]. Carbonyl groups, such as aldehydes and ketones, have proven to be effective warheads in new covalent inhibitors of SARS-CoV-2 M^pro^. In detail, these groups form a reversible hemithioacetal adduct by reacting with the cysteine–SH group through nucleophilic addition, mimicking the intermediate produced in the enzyme’s natural catalytic cycle and ensuring high stability and extended duration of the inhibitor–protein complex. For example, Zhang et al. [113] developed an α-ketoamide inhibitor for SARS-CoV-2, based on the three-dimensional crystal structure of its M^pro^ protease. This inhibitor, designated as **13b**, binds to the active site of M^pro^, forming hydrogen bonds with the backbone amides of key residues in the oxyanion hole and creating a reversible thiohemiketal covalent bond with the Sγ atom of the catalytic residue Cys145. Compound **13b** inhibits the purified recombinant protease with an IC_50_ of 0.67 ± 0.18 μM and inhibits SARS-CoV-2 replication in Calu3 human lung cells with an EC_50_ of 4–5 μM. At a concentration of ~20 μM, virus replication was inhibited approximately 100-fold.

Numerous excellent articles have discussed the SARS-CoV-2 M^pro^ inhibitors discovered to date [64,106,114,115,116,117,118,119], providing insights into their function, structure, mechanism of inhibition, including structure–activity relationships, protein–inhibitor interactions, and the status of clinical trials. The main types of SARS-CoV-2 M^pro^ inhibitors and relevant examples are summarized in Table 2 and Figure 4 and discussed in the following paragraphs (for a review on the topic with more examples see [64]).

As illustrated in Table 2 and recently reviewed by Li et al. [64], a variety of peptidic, peptidomimetic, and small molecules have been tested as inhibitors of SARS-CoV-2 Mpro. Initial efforts in identifying SARS-CoV-2’s M^pro^ inhibitors involved several covalent warheads, such as aldehydes, nitriles, and others. When these warheads are appropriately positioned within the M^pro^-binding site, they form covalent bonds, usually with the catalytic Cys145. This mechanism involves binding the inhibitor to the active site, where the covalent warhead reacts with the thiol group of Cys145, resulting in enzyme inhibition by blocking its activity. Examples include the early N3 inhibitor and Pfizer’s oral antiviral PF-07321332 (Paxlovid), which employs a nitrile warhead. Covalent inhibitors offer high potency and selectivity but also pose challenges, such as potential toxicity and resistance due to off-target effects and viral mutations. Despite these challenges, covalent warheads remain a powerful strategy in designing inhibitors for SARS-CoV-2’s M^pro^.

A representative M^pro^ covalent peptidomimetic inhibitor with an aldehyde “warhead” is compound **11a** (Figure 4) [63], which demonstrated potent enzymatic activity, with an IC_50_ value of 53 nM and a cellular antiviral EC_50_ value of 530 nM. Although aldehyde-based inhibitors of M^pro^ have potent enzyme activities, the aldehyde group is chemically reactive and often associated with off-target effects and undesired toxicities. Less electrophilic ketones have been explored as the “warhead” group of M^pro^ inhibitors. An example is benzothiazolyl ketone 5h/YH-53 (Figure 4), which is a potent SARS-CoV-2 M^pro^ inhibitor (K_i_ = 18 nM) with an antiviral activity of EC_50_ = 0.15 μM [120]. α, β-Unsaturated esters, amides, and related groups can covalently bind Cys145 through a Michael addition reaction and are, therefore, good “warhead” groups for cysteine proteases. A representative example is inhibitor N3, with a K_i_ of 9.0 μM [131]. Likewise, acrylamide or vinyl sulfone can undergo a Michael addition reaction and be potential warheads; however, most of these types of inhibitors tested exhibited low antiviral activities [64]. Several nitrile-based inhibitors were also tested as SARS-CoV-2 main protease inhibitors, including nirmatrelvir, which showed highly potent activity (K_i_ = 3.11 nM). Oral administration of nirmatrelvir significantly reduced the viral loads in SARS-CoV-2-infected mice [104].

Non-peptidic inhibitors SARS-CoV-2 M^pro^ have also been discovered and developed, with the majority initially identified from compound screening, including virtual screening. Among the flavonoids and related analogs tested, baicalin was identified as an inhibitor of SARS-CoV-2 M^pro^ with an IC_50_ of 6.41 μM, which inhibited replication of SARS-CoV-2 in cells (EC_50_ = 27.87 μM) [122]. Quinoline analogs were also tested, with MAT-POS-e194df51-1 identified as a potent SARS-CoV-2 M^pro^ inhibitor (IC_50_ = 36.8 nM), exhibiting potent anti-SARS-CoV-2 activities with EC_50_ as low as 63.8 nM [132]. Several terpenoid compounds, including Bardoxolone methyl, were found to inhibit SARS-CoV-2 M^pro^ with micromolar IC_50_ values and potent cellular antiviral activity (EC_50_ = 0.29 μM) [124]. Furthermore, activated esters that could covalently bind to Cys145, including pyridinyl esters, benzotriazole esters, and their analogs, were examined as SARS-CoV and SARS-CoV-2 M^pro^ inhibitors. Pyridinyl ester compound GRL-0920 was found to be a potent inhibitor of SARS-CoV-2 M^pro^ [125]. Selenium-containing compounds, such as ebselen and its analog, MR6-18-4, were potent inhibitors of SARS-CoV-2 M^pro^, with the latter exhibiting an IC50 of 0.35 μM and cellular anti-SARS-CoV-2 EC_50_ of 3.74 μM [133]. Benzotriazole-based inhibitors, including ML300, were also identified as SARS-CoV-2 M^pro^ inhibitors, with an IC_50_ value of 4.99 μM [126]. Other classes of non-peptide inhibitors include pyrimidine analogs (e.g., carmofur) [102], compounds containing an acrylamide group (e.g., compound **1e**) [127], isatin analogs (e.g., compound **5f**) [128], metal-containing inhibitors (e.g., JMF1586) [129], and triazine compounds (e.g., S-217622) were tested as SARS-CoV-2 M^pro^ inhibitors, as summarized in Table 2 (the structures of these compounds are presented in Figure 4). Several other miscellaneous compounds including FDA-approved drugs were also tested as SARS-CoV-2 M^pro^ inhibitors (for a review on the topic, see [115]). For example, during the screening of more than 10,000 compounds, including approved drugs, drug candidates in clinical trials, and other pharmacologically active compounds, a structure-based virtual screening combined with a fluorescence energy transfer (FRET)-based assay revealed seven compounds that inhibited M^pro^, with IC_50_ values ranging from 0.67 to 21.4 μM [115]. Two of these compounds, disulfiram and carmofur, are US Food and Drug Administration (FDA)-approved drugs, while ebselen, tideglusib, shikonin, PX-12, and TDZD-8 are in clinical trials or preclinical development. Using molecular dynamics (MD) simulations and docking, Alamri et al. identified three potential inhibitors of SARS-CoV-2 from an integrated library of 1000 molecules and 16 approved protease inhibitors. Compound 621 emerged as the best candidate, although no poses showing thiohemiacetal formation were observed, unlike in previous studies with similar functional groups [134].

Based on a literature survey of FDA-approved drugs with antiviral and antibacterial properties, Pathak et al. [135] screened compounds such as ciclesonide, rifampicin, reserpine, loperamide, elvitegravir, brivudine, pentoxifylline, eugenol, isoniazid, tinidazole, diethylcarbamazine, and vancomycin using docking studies against SARS-CoV-2 M^pro^. Rifampicin and ciclesonide exhibited the highest binding affinity, with rifampicin was identified as the most promising drug, due to its favorable binding energy.

Another study by Kandeel et al. [136], based on virtual screening for repurposing FDA-approved drugs, suggested that a combination of ribavirin, telbivudine, vitamin B12, and nicotinamide could be used for COVID-19 treatment. Additionally, computational synergistic studies by Muralidharan et al. found that a combination of lopinavir, ritonavir, and oseltamivir displayed higher binding affinity to SARS-CoV-2 M^pro^ than each drug individually, indicating potential for repurposing against COVID-19 [137]. However, ribavirin and telbivudine, being nucleoside analogs, are less likely to be effective protease inhibitors. In contrast, the antiretroviral protease inhibitors ritonavir and lopinavir merit further investigation as SARS-CoV-2 M^pro^ inhibitors. It should be noted, however, that a randomized clinical trial involving hospitalized adult patients with confirmed SARS-CoV-2 infection showed no benefit from the lopinavir–ritonavir combination treatment beyond standard care [138].

Several studies have highlighted the potential of phytochemicals from medicinal plants as inhibitors of SARS-CoV-2 M^pro^. For example, Hossain et al. [139] screened a phytochemical library consisting of 2431 compounds from 104 Korean medicinal plants known for their medicinal and antioxidant properties. The library was initially screened using molecular docking, followed by revalidation with a deep learning method. A Recurrent Neural Network (RNN) computing system was employed to develop an inhibitory predictive model using the SARS coronavirus M^pro^ dataset. This model was used to screen the top 12 compounds based on their binding affinity. The top 12 compounds identified were Catechin gallate, Cynaroside, Cosmosiin, Isoquercitrin, Rutin, Hyperoside, Isochlorogenic acid b, Quercetin 3-O-malonylglucoside, Cacticin, Narcissoside, Guaijaverin, and Luteolin-7-O-Rutinoside, all of which exhibited binding affinities ranging from −8.0 to −8.9 kcal/mol. The top two lead compounds, Catechin gallate and Quercetin 3-O-malonylglucoside, were selected based on their inhibitory potency against M^pro^.

In another study, Patel et al. [140] conducted a computational investigation to identify Mpro inhibitors from a library of natural compounds with proven antiviral activities. Using a hierarchical workflow of molecular docking, ADMET assessment, dynamic simulations, and binding free-energy calculations, they identified five natural compounds—Withanosides V and VI, Racemosides A and B, and Shatavarin IX—that exhibited strong binding affinity and stable interactions with Mpro key pocket residues.

Similarly, Mahmud et al. [141] constructed a phytochemical dataset through an extensive literature review and explored the binding potential of various phytochemicals with the main protease using molecular docking. The top three hit compounds, medicagol, faradiol, and flavanthrin, demonstrated binding scores of −8.3, −8.6, and −8.8 kcal/mol, respectively. These compounds bind to the active groove of M^pro^, consisting of residues His41, Cys145, Met165, Met49, Gln189, Thr24, and Thr190, leading to the inhibition of the main protease. The screening of a library of 32,297 phytochemicals and Chinese medicinal agents with potential antiviral properties against SARS-CoV-2 M^pro^ identified 5,7,3′,4′-tetrahydroxy-2′-(3,3-dimethylallyl) isoflavone, myricitrin, and methyl rosmarinate as the most promising candidates [142]. Another investigation into the activity of FDA-approved drugs against SARS-CoV-2 M^pro^ highlighted sincalide, ritonavir, phytonadione, and pentagastrin as potential inhibitors [143]. Shamsi et al. [144] performed virtual screening of a library of 2388 FDA-approved drugs against SARS-CoV-2 M^pro^. From the top ten hits, the antiviral drugs glecaprevir and maraviroc demonstrated the highest binding affinity and effectively bound to the conserved residues in the active site of M^pro^ [144]. Additionally, Gurung et al. [145] screened a library of phytochemicals with previously reported antiviral properties against SARS-CoV-2 M^pro^ using a computational approach, and they identified bonducellpin-D as the most promising lead molecule.

Overall, the extensive efforts and ongoing research following the COVID-19 pandemic have significantly advanced the discovery of antiviral drugs against SARS-CoV-2, including M^pro^ inhibitors through both in silico and in vitro studies. The urgent therapeutic need to combat COVID-19 has underscored the importance of pharmaceutical repurposing (discussed further below) and structure-based drug design. These strategies are crucial for the rapid development of potent drugs, which not only address immediate treatment needs but also save time and resources. Effective structure-based drug design hinges on the availability of high-quality structural data. The ideal inhibitors should exhibit high binding specificity to their target (minimizing off-target effects), competitive binding affinity (enhanced efficacy), flexibility (increased effectiveness), ease of administration, and an acceptable plasma half-life. Despite numerous in silico studies conducted to identify compounds with potential antiviral activity against SARS-CoV-2, very few have been biologically validated.

## 5. Repurposing Commercial Protease Inhibitors for Treating Viral Respiratory Infections

The rapid growth of COVID-19 cases has resulted in a rising death toll and significant disruption to the global economy [146]. De novo drug discovery can take years to progress from concept to market, while drug repurposing offers a more immediate solution [147]. One approach employed to find SARS-CoV-2 M^pro^ inhibitors, particularly at the onset of the pandemic, was drug repositioning [148]. Given the lengthy and costly nature of drug discovery and development, repurposing existing approved (by regulatory authorities) drugs or those already in human clinical trials is a practical approach [149]. This approach involves identifying drugs approved for one disease (with known safety profiles and potential adverse effects) that can be repurposed to treat another condition—in this case, COVID-19 [150].

A widely used computational tool for repositioning drugs or identifying compounds with new activities is protein–ligand docking. This tool predicts whether, and how, a particular molecule can bind to a specific target, such as the SARS-CoV-2 M^pro^ [151]. However, protein–ligand docking has limitations, including treating the protein as a rigid body and the unreliable accuracy of scoring functions in estimating binding energies [152]. Additionally, the flexibility of the SARS-CoV-2 M^pro^ poses a challenge for small-molecule inhibitor design [153]. Furthermore, the key question was whether these readily available drugs can bind to the active site of SARS-CoV-2 proteases to inhibit their proteolytic activity. If successful, these drugs could inhibit SARS-CoV-2 replication and serve as starting points for rational drug design to develop optimal inhibitors of M^pro^, thereby further controlling the virus.

Considering the structural similarities between, HCV NS3/4A protease [154], and SARS-CoV-2 M^pro^, it has been hypothesized that existing successful drugs targeting HCV protease might also function as antivirals against SARS-CoV-2. This approach leverages the efficacy of several approved HCV drugs, potentially accelerating the availability of effective treatments for COVID-19. Although there are structural differences between SARS-CoV-2 M^pro^ and HIV protease, FDA-approved inhibitors of the latter have been tested against M^pro^ (discussed further below). Despite these advancements, the potential of these inhibitors to inhibit HRV–3C^pro^ and AVP has not been extensively studied. This gap in research indicates the need for broader investigations to explore the efficacy of these inhibitors across different viral proteases, which could lead to more comprehensive antiviral strategies.

### 5.1. Repurposing of HIV Protease Inhibitors to Treat Viral Infections of the Respiratory Tract

Protease inhibitors, particularly those targeting HIV, have been explored for their potential cross-reactivity against coronaviruses such as SARS-CoV and SARS-CoV-2 [155]. Initial studies have shown that HIV protease inhibitors might impede the replication of these CoVs. It should be noted that SARS-CoV and SARS-CoV-2 share a 96% amino acid sequence similarity and similar activity pockets [144,156,157]. Notably, Lopinavir/Ritonavir, a combination widely used in HIV treatment, was extensively tested but showed limited efficacy in significant clinical trials against COVID-19 [158,159,160].

Molecular dynamics simulations and other experimental approaches have identified several potential HIV protease inhibitors as candidates against SARS-CoV-2. Despite differences in the structures of M^pro^ and HIV protease, in vitro data indicated virus suppression abilities that may lay a foundation for developing new anti-SARS-CoV-2 small-molecule inhibitors for clinical applications [158]. Among the inhibitors studied, nelfinavir, darunavir, and ritonavir showed some potential in early studies, but further research and clinical trials are required to confirm their effectiveness [161,162,163].

Additionally, Cardoso and Mendanha [155] investigated ten approved HIV protease inhibitors [164], including amprenavir, atazanavir, darunavir, fosamprenavir, indinavir, lopinavir, nelfinavir, ritonavir, saquinavir, and tipranavir, as repurposed drug candidates against SARS-CoV-2 (Table 3 and Figure 5). Despite molecular differences, all exhibited similar behavior in molecular dynamics simulations, highlighting their potential as therapeutic options against SARS-CoV-2 [155].

Moreover, the therapeutic potential of HIV protease inhibitors extends beyond HIV and coronaviruses. Nelfinavir, for instance, was identified as a potent inhibitor of human adenovirus in a novel microscopy-based antiviral screen. While it did not affect genome replication or early gene expression, nelfinavir significantly reduced subsequent rounds of infection by disrupting post-translational modifications crucial for viral progeny production and cell lysis [160,165,166].

Most research efforts to repurpose existing viral protease inhibitors, specifically HIV protease inhibitors, have focused on targeting SARS-CoV-2 due to the urgent global health crisis posed by COVID-19. Despite promising findings, the efficacy of these inhibitors against other viruses, such as HRV–3C^pro^ and AVP, remains underexplored. Preliminary studies are scarce, necessitating a comprehensive evaluation of these inhibitors’ effectiveness against HRV–3C^pro^ and AVP. Consequently, their potential cross-activity against these enzymes has not been extensively studied. Further research is crucial to determine their efficacy in these contexts

To address this gap, in this work, an in silico binding affinity assessment of the ten approved HIV protease inhibitors listed in Table 3 (and Figure 5) [155] against HRV–3C^pro^ and AVP was conducted. The binding affinity of these inhibitors was assessed using PyRx v1.1 (AutoDock Vina). The 3D structures of the target proteins were obtained from the Protein Data Bank (HRV–3C^pro^ PDB: 2b0f and AVP, PDB: 4 pie), and Chimera software v1.7.1 [167] was used to prepare the proteins for molecular docking. The ligand structures were obtained via the PubChem Database [168].

To examine the binding potential of the ten HIV protease inhibitors on HRV–3C^pro^, a blind docking was performed on the entire surface of the protein using PyRx v1.1 (AutoDock Vina), with a box center at (294, 0.645, 0.237) and size (40, 40, 40). Interestingly, all ten compounds were placed in the catalytic site of the protein. Among the ten inhibitors, three exhibited binding affinities lower than −6.5 kcal/mol for HRV–3C^pro^: Indinavir (−7.4 kcal/mol), Saquinavir (−6.7 kcal/mol), and Tipranavir (−7.2 kcal/mol). The interactions of these three compounds with the amino acids of the active site of HRV–3C^pro^ are illustrated in Figure 6. These three compounds fit perfectly into the active site of the enzyme and interact with the amino acids of the catalytic site, including those of the catalytic triad, primarily through van der Waals forces, hydrogen bonding, and Pi-cation interactions.

Likewise, a blind docking of the compounds of Table 3 (and Figure 5) was performed on the entire surface of AVP using the AutoDock Vina, with box center (11, −3,–13) and size (40, 40, 40). Out of ten compounds, eight (except Atazanavir and Saquinavir) were placed close to the active site and redocking with box center (8.9, −1.9, −5.0) and size (22, 22, 22). The binding affinity of the eight compounds are presented in Table 3, and as shown, six of them (Amprenavir, Darunavir, Indinavir, Lopinavir, Nelfinavir, and Tipranavir) exhibited binding affinities lower than −6.5 kcal/mol. The interaction of these compounds with the catalytic site of AVP are illustrated in Figure 7. As shown, these compounds interact with the amino acids of the AVP primarily through van der Waals interactions and hydrogen bonding. In accordance with previous studies [160,165,166], in this work, the potential of Nelfinavir as a drug to combat infections caused by adenovirus is further highlighted.

### 5.2. Repurposing of HCV NS3/NS4A Protease Inhibitors to Treat Viral Infections of the Respiratory Tract

The importance and potential efficacy of non-structural NS3/4A protease inhibitors of HCV against SARS-CoV-2 have garnered significant attention in the scientific community. It has been suggested that these inhibitors are effective against SARS-CoV-2 [169]. Among the promising candidates are α-ketoamide-containing covalent inhibitors, which are particularly effective at binding to cysteine proteases like M^pro^ [63]. The presence of two adjacent C=O groups creates a powerful electrophile, enhancing their inhibitory action. Boceprevir was one of the first HCV antiviral agents to demonstrate inhibitory activity against both M^pro^ and coronaviruses [170,171,172]. Given the structural similarities between the HCV NS3/4A protease and SARS-CoV-2 M^pro^, several studies suggest that a range of NS3/4A inhibitors could be effective against the SARS-CoV-2 main protease [154]. The HCV NS3/4A protease inhibitors that have been considered include FDA-approved drugs such as boceprevir, telaprevir, ritonavir, asunaprevir, paritaprevir, grazoprevir, and glecaprevir (Table 4 and Figure 8) [173].

Similar to known HIV protease inhibitors, the inhibitory effects of HCV NS3/NS4A protease inhibitors on both HRV–3C^pro^ and AVP have not been extensively studied. To address this gap, docking studies of the 11 known HCV NS3/NS4A protease inhibitors on HRV–3C^pro^ were initially performed. As described previously, a blind docking on the entire surface of the protein using AutoDock Vina, with box center at (294, 0.645, 0.237) and a size of (40, 40, 40), was conducted. Notably, only 3 out of the 11 compounds—Simeprevir, Telaprevir, and Vaniprevir—were positioned close to the catalytic site. Subsequently, these three compounds were redocked with a box center at (296, −8.58, 4.8) and a size of (20, 20, 20). The binding affinities of these compounds are presented in Table 4, with Simeprevir exhibiting the lowest binding affinity (−8.2 kcal/mol). The interactions of these compounds with the catalytic site of HRV–3C^pro^ are illustrated as 2D images in Figure 9. As shown, the three compounds could interact with the amino acids of the catalytic triad, particularly with His40 and Glu71, through electrostatic (van der Waals) interactions. Nevertheless, further in silico and in vitro experiments are required to verify the inhibitory potential against HRV–3C^pro^.

Subsequently, binding studies of the 11 known HCV NS3/NS4A protease inhibitors (Figure 8) on AVP were performed. The initial blind docking on the entire surface of the protein with box center (11, −3, −13) and size (40, 40, 40) placed 6 compounds, namely Danoprevir, Grazoprevir, Paritaprevir, Simeprevir, Telaprevir, and Vaniprevir, close to the active site. These compounds were redocked with box center (8.9, −1.9, −5.0) and size (22, 22, 22). The binding affinities of the eight compounds are presented in Table 4, and as shown, all of them exhibited values less than −7.5 kcal/mol. Paritaprevir exhibited the lowest binding affinity (−10.1 kcal/mol), followed by Simeprevir (−8.5 kcal/mol). Notably, the binding affinities of these compounds against SARS-CoV-2 M^pro^ were reported to be approximately −10.7 kcal/mol, highlighting the potential of both compounds to be used as leads for the development of dual/multiple inhibitors for the treatment of infections caused by viruses, including SARS-CoV-2 and adenovirus. The interactions of these compounds with the catalytic site of AVP are illustrated in Figure 10. As shown, these compounds interact with the amino acids of the AVP, including those of the catalytic triad, primarily through van der Waals interactions. Together, the above results indicate that drug repurposing could be an alternative strategy to identify drugs to fight viral infections, including those of the respiratory tract.

## 6. Conclusions and Suggestions

The development of effective therapeutic strategies is crucial in managing VRTIs. Among various approaches, targeting viral proteases has emerged as a promising avenue due to their critical role in the viral lifecycle. This review highlights the significance of protease inhibitors in combating VRTIs caused by major pathogens, including HRV, AdV, and SARS-CoV-2. Many viruses encode one or more proteases essential for their replication, making these enzymes ideal therapeutic targets. The success in targeting HIV-1 and HCV NS3/4A proteases underscores the potential of this strategy. Given the similarities in the proteolytic processes of these viruses, inhibitors designed for SARS-CoV-2 could potentially be adapted to target the proteases of other respiratory viruses like rhinovirus and adenovirus.

Protease inhibitors have shown considerable efficacy in inhibiting the replication and spread of these viruses, thus mitigating the severity of infections. The ongoing COVID-19 pandemic has underscored the importance of rapid and effective antiviral therapies, with protease inhibitors playing a pivotal role in reducing disease impact. This experience provides a valuable foundation for repurposing existing drugs and developing new inhibitors for other respiratory viruses. For example, Macip et al. [119] collected SARS-CoV-2 M^pro^ inhibitors from various sources, including bibliographic records, the COVID Moonshot project, and the ChEMBL database. In the study, the 15 most potent covalent and 15 most potent non-covalent inhibitors of SARS-CoV-2 M^pro^ were identified. Furthermore, well-studied SARS-CoV-2 or SARS-CoV inhibitors, such as N3, Nirmatrelvir, and Ebselen, could be examined further and repurposed for developing novel dual or triple inhibitors. The main question raised here is whether these SARS-CoV-2 M^pro^ inhibitors could be repurposed to treat infections of the respiratory tract caused by other viruses, including those responsible for human rhinovirus and adenovirus infections. Nevertheless, studies in assessing the efficiency of these inhibitors against HRV–3C^pro^ are currently in progress in our laboratories

The extensive research and clinical trials initiated in response to COVID-19 have greatly enhanced our understanding of antiviral drug development, particularly concerning protease inhibitors. Drugs like lopinavir/ritonavir and darunavir, initially developed for HIV, were quickly repurposed in attempts to combat SARS-CoV-2, demonstrating the potential for rapid drug repurposing. Despite mixed clinical outcomes, these efforts highlight the feasibility of redirecting existing therapeutic agents against novel viral targets. Future efforts should prioritize the rational design of novel inhibitors, leveraging computational modeling and high-throughput screening technologies used during the pandemic. Additionally, interdisciplinary collaboration will be essential to translate these findings into clinical applications, ensuring that new treatments are safe, effective, and accessible.

The potential cross-activity of SARS-CoV-2 protease inhibitors against other viral proteases remains largely unexplored and represents a significant opportunity for research. Recent studies, like those by Liu et al. [174], which developed dual inhibitors effective against both SARS-CoV-2 and HRV–3C^pro^, illustrate the possibilities of such cross-viral therapeutic applications. These results guide the design of inhibitors that are either virus-specific or retain a broad antiviral spectrum.

Despite recent advances in the development of inhibitors targeting the protease of viruses affecting the respiratory tract, the high mutation rate in viral genomes can lead to changes in the protease’s amino acid sequence [175], potentially affecting its structure and function. These mutations can alter protease cleavage sites or the overall structure of the enzyme, impacting the efficacy of protease inhibitors developed to target these viruses resulting in drug resistance [176]. Regarding the protease sites, there is limited to no information on how specific mutations would alter the binding of potential drugs on HRV–3C^pro^ and AVP.

HRVs display considerable genetic diversity, which can lead to mutations in the 3C^pro^. Variations in 3C^pro^ can influence its cleavage specificity and efficiency, potentially leading to resistance against inhibitors designed to target conserved elements of the protease. Although there is no direct evidence showing that certain mutations reduce the binding affinity of inhibitors on HRV–3C^pro^, studies by Leong et al. [177] demonstrated that 3C^pro^ can bind specifically to the 5′-terminal 126 nucleotides of the viral RNA (126 RNA) in addition to efficiently cleaving a synthetic peptide in trans. Single amino acid substitutions at residues highly conserved among picornaviruses or within the putative catalytic triad affected binding and proteolytic activity. For example, the D85N mutation destroyed the ability of 3C^pro^ to bind specifically to the 126 RNA, while substitutions at His40, Glu71, or Cys146 resulted in proteolytically inactive mutants that could still bind to the RNA. This suggests distinct domains in 3C^pro^ for RNA binding and proteolytic activities.

Although AdVs are more genetically stable than RNA viruses [23], mutations in the AVP can still occur and may affect drug interactions. Adenoviruses constantly mutate during circulation in the human population, though related phenotypic changes are rarely detected because of limited studies of emergent strains [178]. Mutations could alter the configuration of the protease’s active site, affecting the effectiveness of antiviral agents [97].

In contrast, there is extensive information on how specific mutations may affect the binding of potential inhibitors to the SARS-CoV-2 M^pro^. The evolution of SARS-CoV-2 has led to multiple variants, some with mutations in the M^pro^, crucial for viral replication [179]. Such mutations can confer resistance to inhibitors that target M^pro^. Changes near the substrate-binding site of M^pro^ could influence the binding of covalent inhibitors, reducing their efficacy. For instance, Hu et al. [180] characterized 102 naturally occurring Mpro mutants located at 12 residues at the nirmatrelvir binding site. Among these, 22 mutations in 5 residues showed comparable enzymatic activity to the wild-type while being resistant to nirmatrelvir. Using recombinant SARS-CoV-2 viruses, they confirmed drug resistance and showed that M^pro^ mutants with reduced enzymatic activity had attenuated viral replication. Jiang et al. [181] expressed six M^pro^ mutants identified in Omicron variants and solved the crystal structures of PF-07304814 bound to these mutants. Structural analysis provided insight into the key molecular determinants responsible for the interaction between PF-07304814 and these mutants of M^pro^. The patterns for PF-07304814 to bind with these M^pro^ mutants and the wild-type M^pro^ are generally similar but with some differences.

The above suggests that viral mutations in proteases present a significant challenge to the development and long-term efficacy of protease inhibitors. This underscores the need for a robust surveillance system to track genetic changes in viral proteases and for designing inhibitors effective against a broad spectrum of viral variants.

In conclusion, the knowledge and resources amassed during the fight against COVID-19 are invaluable for the ongoing and future battles against respiratory viral infections. By advancing our understanding and development of protease inhibitors, informed by our recent experiences, we can better prepare for and respond to both current and future viral threats. This review underscores the transformative potential of drug repurposing and novel drug development for the management of respiratory tract viral infections, catalyzed by the unprecedented focus on antiviral research during the COVID-19 pandemic.

## Figures and Tables

**Figure 1 ijms-25-08105-f001:**
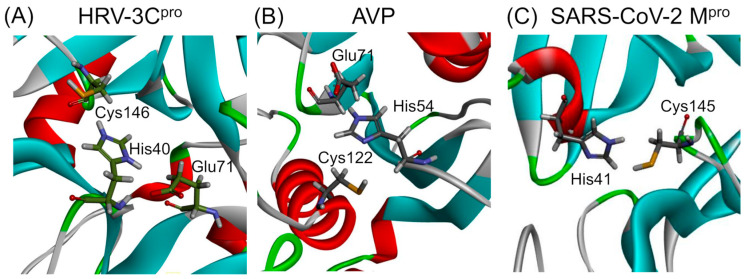
Ribbon diagrams presenting the catalytic sites of proteases from viruses affecting the respiratory tract. The catalytic site of HRV–3C^pro^ (PDB: 2b0f) consists of a His40–Glu71–Cys146 triad, while that of AVP (PDB: 4 pie) consists of a His54–Glu71–Cys122 catalytic triad. In contrast, the SARS-CoV M^pro^ (PDB: 6lu7) features a His41–Cys145 catalytic dyad. The amino acids of the catalytic sites are presented in stick representation. The 3D structures of the three proteases have been modified to highlight the amino acids forming the catalytic sites.

**Figure 2 ijms-25-08105-f002:**
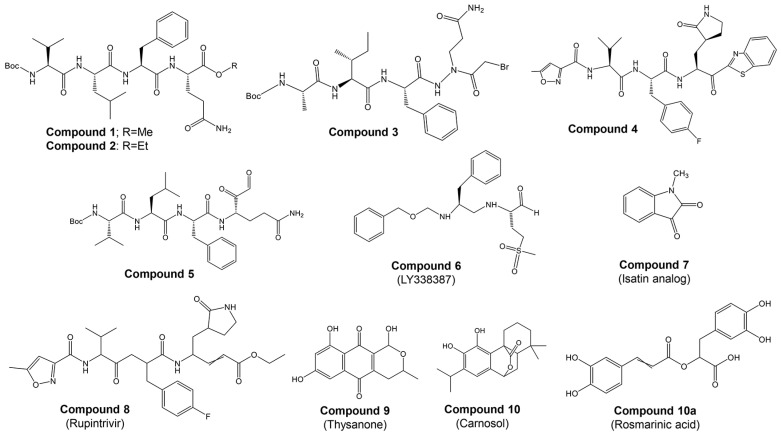
Structures of some examples of known HRV–3C^pro^ inhibitors.

**Figure 3 ijms-25-08105-f003:**
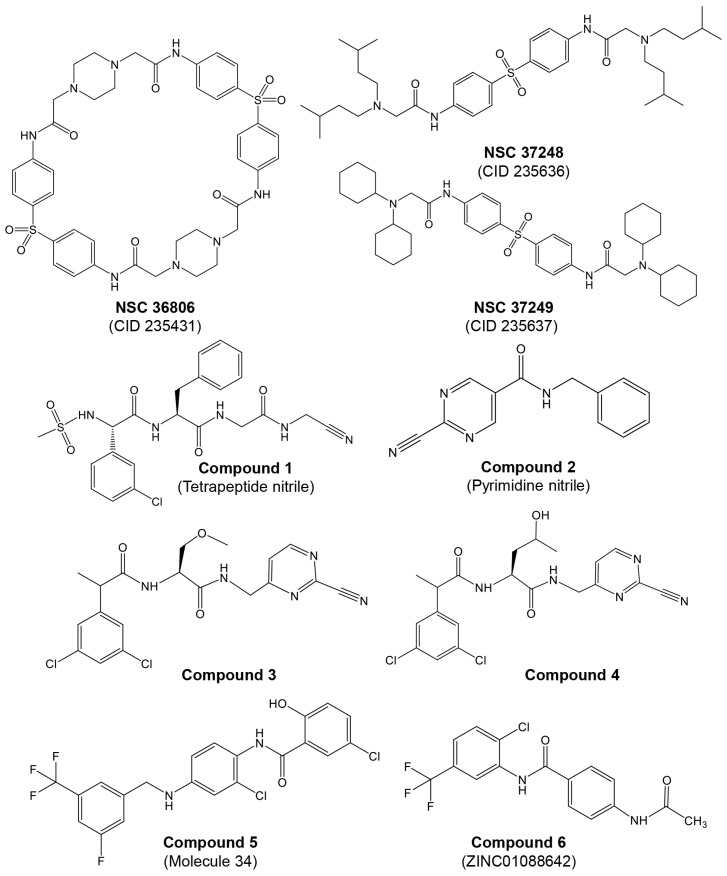
Examples of various classes of adenovirus protease inhibitors.

**Figure 4 ijms-25-08105-f004:**
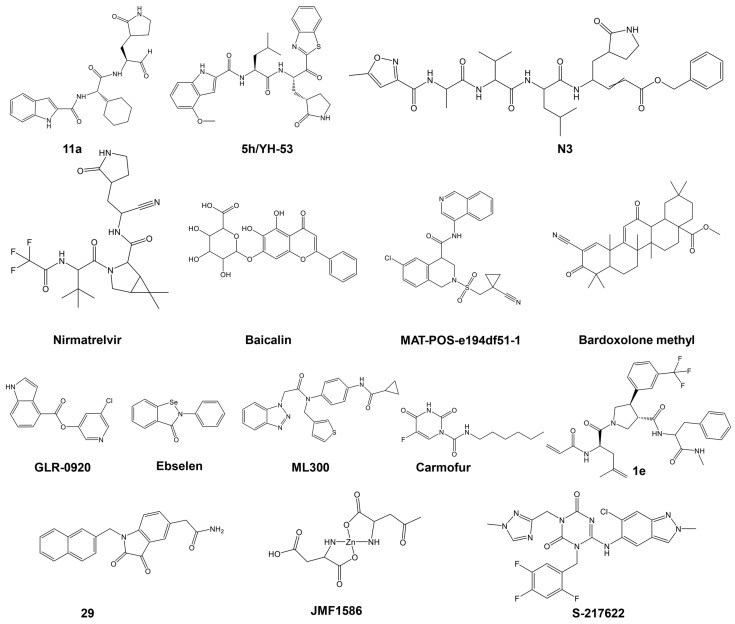
Structures of various classes of SARS-CoV-2 M^pro^ are presented in Table 2.

**Figure 5 ijms-25-08105-f005:**
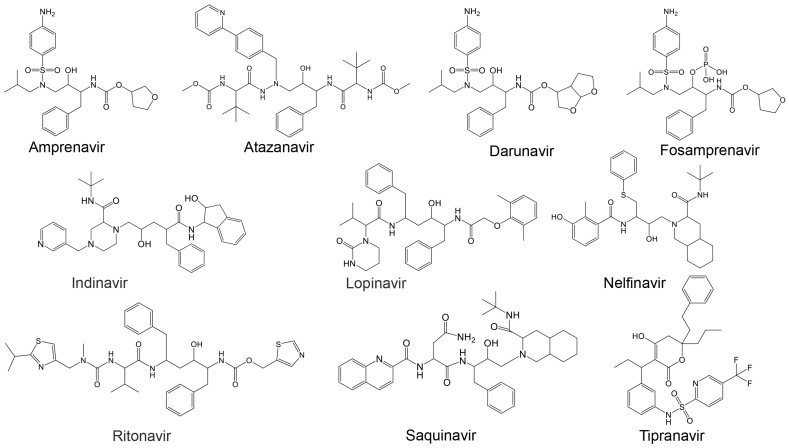
Structures of known HIV protease inhibitors tested against the viral proteases of Table 3.

**Figure 6 ijms-25-08105-f006:**
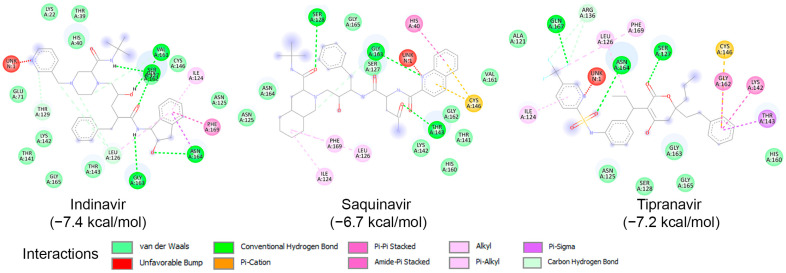
Molecular interactions of HRV–3C^pro^ with three known HIV protease inhibitors namely Indinavir, Saquinavir, and Tipranavir, displayed as 2D images. The binding energy (in kcal/mol) for each inhibitor is indicated below their respective interaction diagrams. The different types of interactions are shown at the bottom of this Figure. The amino acid residues of the HRV–3C^pro^ involved in the interactions are labeled and highlighted in colored circles. These circles correspond to different interactions indicated at the bottom of this Figure. The 3D structure of HRV–3C^pro^ was obtained from the protein data bank (PDB: 2b0f). Docking studies were carried out using PyRx v1.1 (AutoDock Vina).

**Figure 7 ijms-25-08105-f007:**
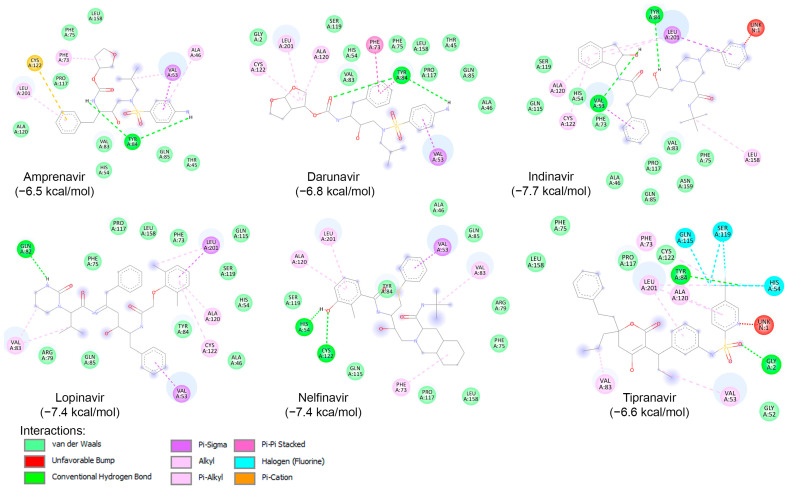
Molecular interactions of AVP with six known HIV protease inhibitors: Amprenavir, Darunavir, Indinavir, Lopinavir, Nelfinavir, and Tipranavir, displayed as 2D images. The binding energy (in kcal/mol) for each inhibitor is indicated below their respective interaction diagrams. The different types of interactions are shown at the bottom of this Figure. The amino acid residues of the adenovirus protease involved in the interactions are labeled and highlighted in colored circles. These circles correspond to different interactions indicated at the bottom of this Figure. The 3D structure of AVP was obtained from the protein data bank (PDB: 4 pie). Docking studies were carried out using PyRx v1.1 (AutoDock Vina).

**Figure 8 ijms-25-08105-f008:**
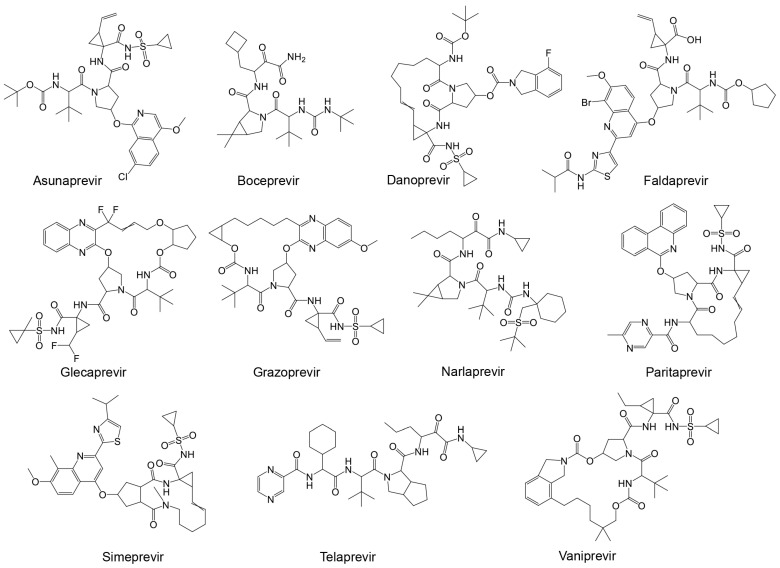
Structures of known HCV NS3/NS4A protease inhibitors tested against the viral proteases presented in Table 4.

**Figure 9 ijms-25-08105-f009:**
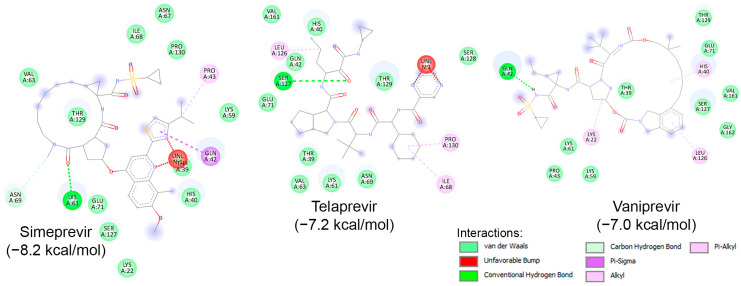
Molecular interactions of HRV–3C^pro^ with three known HCV NS3/NS4A protease inhibitors, namely Simeprevir, Telaprevir, and Vaniprevir, displayed as 2D images. The binding energy (in kcal/mol) for each inhibitor is indicated below their respective interaction diagrams. The different types of interactions are shown at the bottom of this Figure. The amino acid residues of the HRV–3C^pro^ involved in the interactions are labeled and highlighted in colored circles. These circles correspond to different interactions indicated at the bottom of this Figure. The 3D structure of HRV–3C^pro^ was obtained from the protein data bank (PDB: 2b0f). Docking studies were carried out using PyRx v1.1 (AutoDock Vina).

**Figure 10 ijms-25-08105-f010:**
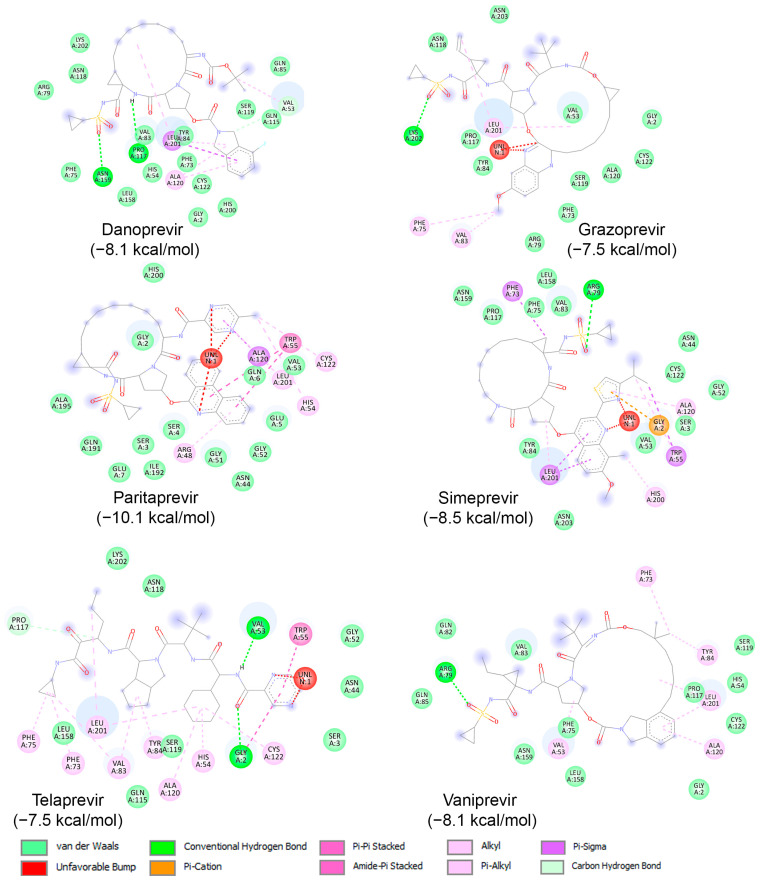
Molecular interactions of AVP with three known HCV NS3/NS4A protease inhibitors, namely Danoprevir, Grazoprevir, Paritaprevir, Simeprevir, Telaprevir, and Vaniprevir, displayed as 2D images. The binding energy (in kcal/mol) for each inhibitor is indicated below their respective interaction diagrams. The different types of interactions are shown at the bottom of this Figure. The amino acid residues of the AVP involved in the interactions are labeled and highlighted in colored circles. These circles correspond to different interactions indicated at the bottom of this Figure. The 3D structure of AVP was obtained from the protein data bank (PDB: 4 pie). Docking studies were carried out using PyRx v1.1 (AutoDock Vina).

**Table 1 ijms-25-08105-t001:** The main characteristics of the virus that are examined in this work.

Virus	Family	Characteristics
Rhinovirus	Picornaviridae	Non-enveloped, single-stranded RNA virus; causes common cold; more than 160 known serotypes.
Adenovirus	Adenoviridae	Non-enveloped, double-stranded DNA virus; causes respiratory illnesses, conjunctivitis, gastroenteritis.
SARS-CoV-2	Coronaviridae	Enveloped, single-stranded RNA virus; causes COVID-19, characterized by fever, cough, respiratory distress

**Table 2 ijms-25-08105-t002:** Examples of various types of SARS-CoV-2 M^pro^ inhibitors.

Inhibitor Type	Sub-Type	Compound/ Example	IC50 (μM)	EC50 (μM)	Notes	Ref.
Peptidic and Peptidomimetic	Aldehyde-based	Compound **11a**	0.053	0.53	Potent enzymatic activity, strong cellular antiviral effects	[63]
	Ketone-based	Benzothiazolyl ketone 5h/YH-53	K_i_ = 17.6 nm	0.15	Potent inhibitor, favorable drug properties	[120]
	α,β-Unsaturated esters/Michael acceptors	N3	K_i_ = 9.0 μΜ	16.77	Inhibits SARS-CoV-2 replication	[102]
	Nitrile-based	Nirmatrelvir	K_i_ = 3.11 nm	74.5	Highly potent, reduced viral loads in mice	[121]
Non-peptidic Small Molecule	Flavonoids	Baicalin	6.41	27.87	Inhibits replication of SARS-CoV-2 in cells	[122]
	Quinoline analogs	MAT-POS-e194df51-1	0.0368	0.0638	Covalent inhibitor Potent anti-SARS-CoV-2 activities	[123]
	Terpenoids	Bardoxolone methyl	5.81	0.29	Potent cellular antiviral activity	[124]
	Pyridinyl ester	GRL-0920	N.A. *	2.8	Blocks cellular viral replication	[125]
	Ebselen analogs	Ebselen	0.67	4.67	Inhibited cellular replication of SARS-CoV-2	[102]
	Benzotriazole-based	ML300	4.99	19.90	Modest antiviral activity in cells	[126]
	Pyrimidine analogs	Carmofur	1.82	N.A.	Non-specific inhibitor	[102]
	Acrylamide and related compounds	Compound **1e**	2.0	33	Weak anti-SARS-CoV-2 activity in cells	[127]
	Isatin analogs	Compound **29**	0.045	N.A.	Probably covalent inhibition	[128]
	Metal-containing inhibitors	JMF1586	N.A.	N.A.	Potent inhibitors, no cellular antiviral activities reported	[129]
	Triazine compounds	S-217622 (Ensitrelvir)	0.013	0.29–0.50	Potent antiviral activities in Vero cells, approved in Japan	[130]

* N.A. not available.

**Table 3 ijms-25-08105-t003:** Binding affinities of HIV protease inhibitors against SARS-CoV-2 M^pro^, HRV–3C^pro^, and AVP.

Name	PubChem CID	Molecular Mass (g/mol)	Binding Affinity (kcal/mol)		
			SARS-CoV-2 M^pro 1^	HRV 3C^pro 2^	AVP ^2^
Amprenavir	65016	505.63	−7.7	−5.9	−6.5
Atazanavir	148192	704.86	−8.8	−6.3	-
Darunavir	213039	547.66	−8.0	−6.2	−6.8
Fosamprenavir	131536	585.61	−7.7	−5.4	−6.2
Indinavir	5362440	613.79	−8.1	−7.4	−7.7
Lopinavir	92727	628.8	−8.4	−6.2	−7.4
Nelfinavir	64143	567.78	−8.3	−6.3	−7.4
Ritonavir	392622	720.94	−7.8	−5.6	−6.4
Saquinavir	441243	670.84	−8.8	−6.7	-
Tipranavir	54682461	505.63	−7.8	−7.2	−6.6

^1^ Obtained from [155]; ^2^ calculated with PyRx v 1.1 (Autodock) as described in the text; the binding affinities of the compounds that were positioned away from the active site during the initial screening are not presented.

**Table 4 ijms-25-08105-t004:** Binding affinities of HVC protease inhibitors against SARS-CoV-2 M^pro^, HRV–3C^pro^, and AVP.

Name	PubChem CID	Molecular Mass (g/mol)	Binding Affinity (kcal/mol)		
			SARS-CoV-2 M^pro 1^	HRV- 3C^pro 2^	AVP ^2^
Asunaprevir	16076883	748.3	−8.19	-	-
Boceprevir	10324367	519.7	−9.44	-	-
Danoprevir	11285588	731.8	−9.99	-	−8.1
Faldaprevir	42601552	869.8	−6.18	-	-
Glecaprevir	66828839	838.9	−9.51	-	-
Grazoprevir	44603531	766.9	−9.71	-	−7.5
Narlaprevir	11857239	708.0	−7.09	-	-
Paritaprevir	45110509	765.9	−10.71	-	−10.1
Simeprevir	24873435	749.9	−10.75	−8.2	−8.5
Telaprevir	3010818	679.8	−11.01	−7.2	−7.5
Vaniprevir	24765256	757.9	−7.56	−7.0	−8.1

^1^ Obtained from [169]; ^2^ calculated with PyRx v.1.1, (Autodock) as described in the text; the binding affinities of the compounds that were positioned away from the active site during the initial screening are not presented.

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
