# Peer review of "Breaking the Chain: Protease Inhibitors as Game Changers in Respiratory Viruses Management"

_ijms, 2024, doi:10.3390/ijms25158105_

Round 1

Reviewer 1 Report

Comments and Suggestions for Authors

The Author proposes a very good survey on protease inhibitors for the management of respiratory viral diseases. The paper is well organised and each section is clearly described and detailed. 

My only inputs are as follows:

- revise the title of chapter 2 in "Respiratory Viruses Depend on proteases for replication"

- the Author should also discuss the impact of  viral mutations on proteases activity for the three virus under investigation. 

Is there the risk that mutations occurring in proteases can affect their cleavage activity and the expected drug efficacy? This important concept should be discussed in the paper main text.

Some information are provided for SARSCoV-2 in the paper "Pozzi C, Vanet A, Francesconi V, Tagliazucchi L, Tassone G, Venturelli A, Spyrakis F, Mazzorana M, Costi MP, Tonelli M. Antitarget, Anti-SARS-CoV-2 Leads, Drugs, and the Drug Discovery-Genetics Alliance Perspective. J Med Chem. 2023 Mar 23;66(6):3664-3702. doi: 10.1021/acs.jmedchem.2c01229. and "Kang BM, Kim D, Kim J, Baek K, Park S, Shin HE, Lee MH, Kim M, Kim S, Lee Y, Kwon HJ. Analysis of SARS-CoV-2 Mutations after Nirmatrelvir Treatment in a Lung Cancer Xenograft Mouse Model. Biomol Ther (Seoul). 2024 Jun 5. doi: 10.4062/biomolther.2023.195."

- Please also cite the following papers

Jensen LM, Walker EJ, Jans DA, Ghildyal R. Proteases of human rhinovirus: role in infection. Methods Mol Biol. 2015;1221:129-41. doi: 10.1007/978-1-4939-1571-2_10. 

Dodge MJ, MacNeil KM, Tessier TM, Weinberg JB, Mymryk JS. Emerging antiviral therapeutics for human adenovirus infection: Recent developments and novel strategies. Antiviral Res. 2021 Apr;188:105034. doi: 10.1016/j.antiviral.2021.105034.

Comments on the Quality of English Language

The English style is good

Reviewer 2 Report

Comments and Suggestions for Authors

Comments to the Authors:

This review emphasizes the importance of identifying and developing inhibitors that target key proteases of major respiratory viruses, including human rhinoviruses (RVs), adenovirus (AdVs), and SARS-CoV-2. By detailing the mechanisms of action and therapeutic potential of these inhibitors, the review aims to demonstrate their significant role in transforming the management of respiratory viral diseases and provide insights into future research directions.

Overall, this data is both interesting and well-written.

Minor suggestion:

  1. In Figures 2, 3, 4, 5, and 8, the authors have represented the structures only. Could you please add the IC50 and EC50 values, if available in the literature, where you cited? Some of these values are mentioned in Table 1. Including them would make the information more informative for readers.
